# Targeting PI3K/AKT/mTOR and MAPK Signaling Pathways in Gastric Cancer

**DOI:** 10.3390/ijms25031848

**Published:** 2024-02-03

**Authors:** Diana-Theodora Morgos, Constantin Stefani, Daniela Miricescu, Maria Greabu, Silviu Stanciu, Silvia Nica, Iulia-Ioana Stanescu-Spinu, Daniela Gabriela Balan, Andra-Elena Balcangiu-Stroescu, Elena-Claudia Coculescu, Dragos-Eugen Georgescu, Remus Iulian Nica

**Affiliations:** 1Discipline of Anatomy, Carol Davila University of Medicine and Pharmacy, 050474 Bucharest, Romania; theodora.morgos@drd.umfcd.ro; 2Department I of Family Medicine and Clinical Base, “Dr. Carol Davila” Central Military Emergency University Hospital, 010825 Bucharest, Romania; 3Discipline of Biochemistry, Faculty of Dentistry, Carol Davila University of Medicine and Pharmacy, 050474 Bucharest, Romania; maria.greabu@umfcd.ro; 4Department of Internal Medicine and Gastroenterology, Carol Davila University of Medicine and Pharmacy, Central Military Emergency University Hospital, 010825 Bucharest, Romania; silviu.stanciu@umfcd.ro; 5Emergency Discipline, University Hospital of Bucharest, 050098 Bucharest, Romania; silvia.nica@umfcd.ro; 6Discipline of Physiology, Faculty of Dentistry, Carol Davila University of Medicine and Pharmacy, 050474 Bucharest, Romania; iulia.stanescu@umfcd.ro (I.-I.S.-S.); daniela.balan@umfcd.ro (D.G.B.); andra.balcangiu@umfcd.ro (A.-E.B.-S.); 7Discipline of Oral Pathology, Faculty of Dentistry, Carol Davila University of Medicine and Pharmacy, 020021 Bucharest, Romania; elena-claudia.coculescu@umfcd.ro; 8Department of General Surgery, Faculty of Medicine, Carol Davila University of Medicine and Pharmacy, 50474 Bucharest, Romania; dragos-eugen.georgescu@umfcd.ro; 9Department of General Surgery, “Dr. Ion Cantacuzino” Clinical Hospital, 020475 Bucharest, Romania; 10Central Military Emergency University Hospital “Dr. Carol Davila”, 010825 Bucharest, Romania; remus.nica@umfcd.ro; 11Discipline of General Surgery, Faculty of Midwifery and Nursing, Carol Davila University of Medicine and Pharmacy, 050474 Bucharest, Romania

**Keywords:** gastric cancer, PI3K/AKT/mTOR, MAPK, gene mutations, bacterial infection, dysregulation, inhibitors

## Abstract

Gastric cancer (GC) is the fourth leading cause of death worldwide, with more than 1 million cases diagnosed every year. *Helicobacter pylori* represents the main risk factor, being responsible for 78% of the cases. Increased amounts of salt, pickled food, red meat, alcohol, smoked food, and refined sugars negatively affect the stomach wall, contributing to GC development. Several gene mutations, including *PIK3CA*, *TP53*, *ARID1A*, *CDH1*, *Ras*, *Raf*, and *ERBB3* are encountered in GC pathogenesis, leading to phosphatidylinositol 3-kinase (PI3K) protein kinase B (AKT)/mammalian target of rapamycin (mTOR)—PI3K/AKT/mTOR—and mitogen-activated protein kinase (MAPK) signaling pathway activation and promoting tumoral activity. *Helicobacter pylori*, growth factors, cytokines, hormones, and oxidative stress also activate both pathways, enhancing GC development. In clinical trials, promising results have come from monoclonal antibodies such as trastuzumab and ramucirumab. Dual inhibitors targeting the PI3K/AKT/mTOR and MAPK signaling pathways were used in vitro studies, also with promising results. The main aim of this review is to present GC incidence and risk factors and the dysregulations of the two protein kinase complexes together with their specific inhibitors.

## 1. Introduction

### 1.1. Gastric Cancer Incidence

Gastric cancer (GC) is a multifunctional disease that is the fourth leading cause of death worldwide [1,2,3,4,5], without any symptoms in early stages [6]. Even though the incidence of GC has decreased since the 1970s due to medical progress, it still represents a major health problem, particularly in Eastern Asian countries, where over 1 million cases were reported in 2020 [7]. GC is considered an aggressive malignancy characterized by uncontrolled cell proliferation and metastasis [8]. Globally, every year, more than 1 million people are diagnosed with GC [9,10].

This malignant pathology is more often found in men, being approximately twice as common compared with women, and usually occurs after 60 years of age [11]. The lower risk of women developing GC is due to estrogen, an effect that lasts until menopause [12]. Moreover, GC is often found in the tropics [13]. GC incidence and mortality vary, with increased rates being observed in Eastern and Western Asia, while in North America, Northern Europe, and most African regions, decreased incidence is reported [14]. The southern region of India has an increased incidence of gastric adenocarcinoma, while South Asia presents decreased incidence [15]. Usually, 90% of GC involves sporadic cases, and 10% present familial aggregation [16]. Australia also has a decreased rate of GC [17]. These variations may be explained by differences in *Helicobacter pylori* (*H. pylori*) genotypes *cagA* and *vacA*; in areas with elevated GC incidence, type *cagA* is predominant [18]. Worldwide, the GC overall 5-year survival rate is about 20%. In China, GC represents the second leading cause of death among women and the third leading cause in men [19]. Japan has reported a survival rate of more than 70% for GC diagnosed in stages I and II. This elevated survival rate may be due to the mass screening programs adopted [20], where endoscopy is the gold standard method [21]. In the last five decades, the US has reported a decreased incidence of GC, while a subtype of GC, non-cardia, is increasing among adults younger than 50 years of age [22].

### 1.2. Gastric Cancer Classification

The Cancer Genome Atlas (TCGA) classifies GC into four subtypes [23]. Anatomic gastric adenocarcinoma is classified into two subtypes: cardia, which includes the upper stomach adjoining the esophagus, and non-cardia, which involves the mid and distal stomach [24]. Histologically, gastric adenocarcinomas represent 90% of GC and have been subdivided into two main types: the intestinal type, or well-differentiated, and the diffuse type, or undifferentiated [25]. The cardia subtype represents one-quarter of cases, while non-cardia represents three-quarters of the incidence worldwide [26]. According to the Laurean classification, the majority of GC cases are represented by adenocarcinoma (90%), and 10% are lymphoma or gastrointestinal stromal tumors [27]. The intestinal type accounts for 50% of cases, the diffuse type 33%, and 17% of cases are mixed or unclassified [27].

The World Health Organization (WHO) classifies GC into four subtypes: papillary, tubular, mucinous, and poorly cohesive [28]. Moreover, Epstein–Barr virus (EBV)-associated GC is another GC subtype, with definitive clinical and molecular characteristics [29], together with human epithelial growth receptor 2 (HER2), which is overexpressed in 6–30% of GC cases [30]. Infection with EBV contributes to 9% of all GC [31]. HER2 is also overexpressed in other malignant pathologies, including breast cancer [32]. The TCGA analyzed 295 primary gastric adenocarcinomas and further proposed a new classification of GC into four subtypes, including EBV-positive, microsatellite instability (MSI), genomic stability (GS), and chromosomal instability (CIN). The Asian Cancer Research Group (ACRG) also provides another classification of the four mentioned GC subtypes, as follows: (i) MSI, (ii) epithelial-to-mesenchymal transition (EMT)/GC, (iii) microsatellite stable with intact *TP53* activity (MSS/*TP53+*), and (iv) microsatellite stable GC with loss of *TP53* activity (MSS/*TP53*−) [33,34]. Moreover, GC intestinal histology is often found in Caucasians.

The cardia subtype is less detected in Africa and Latin America [35]. With the lack of significant symptoms in the early stages, most GC cases are diagnosed in middle or late stages, leading to a decreased survival rate [36,37]. In the early stages, a small number of patients manifest nausea, vomiting, or symptoms similar to ulcers. In advanced GC stages, patients usually report pain and weight loss as the most common symptoms [38]. Unfortunately, the 5-year survival rate is about 20–30% for advanced GC cases [39].

## 2. Risk Factors

### 2.1. Helicobacter pylori

*H. pylori* infection represents the main risk factor involved in GC pathology, responsible for 78% of cases [40]. *H. pylori* is a Gram-negative bacterium that leads to infection during childhood in 50% of the global population [41]. The infection with *H*. *pylori* leads to chronic gastritis and further induces the development of atrophic gastritis, dysplasia, intestinal metaplasia, and finally GC [42,43]. Atrophy, intestinal metaplasia, and low-grade and high-grade dysplasia of the stomach mucosa are recognized as premalignant lesions for GC development [44].

According to the WHO, *H. pylori* infection is considered as the definitive gastric carcinogen linked with crowded living conditions, family members who share the same bedroom, low socioeconomic status, low educational level, poor sanitation conditions, and not washing hands before meals [45]. Moreover, it is considered that *H. pylori* infection is acquired in childhood and is usually asymptomatic, except in the cases that develop peptic ulcer disease [46]. In 2013, Japan implemented an eradication strategy against *H. pylori*, but the GC incidence did not decrease substantially because the gastric microbiome plays an important role in GC development [47]. *H. pylori* is a risk factor, especially for non-cardia GC [48]. Gastrointestinal endoscopy plays an important role in GC detection and may reduce patient mortality [49]. Therefore, in countries with elevated cases of GC, endoscopic and radiographic screening for the asymptomatic general population has been adopted as a prevention strategy [50].

In gastric epithelial cells, *H. pylori* induces overexpression of metalloproteinases (MMPs) MMP-1, 3, 7, 9, and 10 [51]. The tumor microenvironment (TME) contains tumor cells and non-tumor cells that are involved in tumor progression, such as stromal cells. In addition, tumor-associated macrophages (TAMs) are the main TME-immune cells that play key roles in cancer progression. Therefore, TAMPs are heterogeneous cells classified as activated M1 phenotype and M2 phenotype [52]. The TME of solid tumors is a hypoxic environment, where ATP suffers hydrolysis and produces a huge amount of adenosine [53]. EMT facilitates the polarized epithelial cell to obtain a mesenchymal phenotype that allows the tumor cells to intravasate into blood or lymph vessels and further develop distant metastasis [54]. Therefore, the risk factors will induce a proinflammatory state, contributing to GC development [55].

### 2.2. Diet

Apart from *H. pylori* and EBV infection, other risk factors can influence the development of GC, such as diet, lifestyle, and metabolic factors [35]. According to World Cancer Research, excessive salt consumption may act as a gastric mucosa stimulant, leading to atrophic gastritis, elevated DNA synthesis, and cell proliferation, increasing the GC risk [56], including for adenocarcinoma [57]. 

In addition, elevated salt levels disturb the gastric mucosal defense barrier, promoting *H. pylori* colonization and allowing entry of carcinogenic compounds [58]. Besides high salt intake, other foods with a negative impact on the stomach are pickled foods, dried fish, meat, and refined carbohydrates [59]. Salty and spicy foods may irritate the stomach wall and may contribute to GC development [60]. Yuan and colleagues conducted a study that included 180 patients with advanced GC and reported that eating hot food and pickled vegetables was correlated with an increased risk of GC, while the consumption of onion or garlic, drinking tea, and eating fresh fruits had protective effects against GC [61]. 

Furthermore, obesity, meat, alcohol consumption, smoked food, and smoking are considered to promote carcinogenesis in the stomach [62,63]. Obesity may contribute to proximal GC development [64] through several mechanisms, including obesity-associated gastro-esophageal reflux, insulin resistance, dysregulation of the levels of leptin, adiponectin, and ghrelin, and elevated levels of insulin-like growth factor (IGF) [65]. Decreased high-density lipoprotein cholesterol (HDL-C) and increased low-density lipoprotein cholesterol (LDL-C) levels are considered GC risk factors. Decreased HDL-C levels in patients with GC are explained by the development of HDL receptors used to prevent intracellular cholesterol accumulation during neoplastic development. In cancer, HDL-C has antioxidant and anti-inflammatory properties [66].

The Mediterranean diet has been shown to have a protective effect [67], with the daily consumption of grains, fruits, olive oil, fish, nuts, and vegetables seemingly inhibiting tumor angiogenesis and reducing DNA damage [68]. However, the results of several meta-analyses concluded that body mass index (BMI) and GC may not have a positive association [69]. The results of a large meta-analysis study that included subjects from 24 prospective studies revealed that neither being overweight nor obesity were associated with an increased risk of developing GC [70]. The International Agency for Research on Cancer (IARC) reported that a positive correlation between processed meat and an increased risk of GC has been observed [71]. The results of several meta-analyses illustrated that a higher intake of red and processed meat is associated with an increased risk of GC, of 41% and 57%, respectively [72]. It is well known that red meat is rich in hemoglobin, myoglobin, and cytochromes, which undergo heat treatment during cooking and generate free radicals, leading to lipids, proteins, and nucleic acid damage, thus promoting cancer development [73]. On the other hand, the consumption of white meat decreases the incidence of GC by 20% [72].

Ethanol is metabolized into the liver by alcohol dehydrogenase (ADH), catalase, or cytochrome p450 2E1 with the generation of acetaldehyde. The ethanol dehydrogenation into aldehyde is coupled with NAD^+^ conversion into NADH, which further is reoxidized to NAD^+^, and generation of reactive oxygen species (ROS). ROS accumulation promotes lipid peroxidation, DNA, and protein damage. According to the IARC, acetaldehyde is considered a group 1 carcinogen for humans, which induces DNA damage in the digestive tract [74]. Li and his research team conducted a study that included 54,682 Japanese men and concluded that alcohol consumption was associated with elevated risk of developing GC among men [75]. Therefore, chronic inflammation of the gastrointestinal tract, induced by risk factors, promotes GC development correlated with a worse survival rate [76].

### 2.3. Systemic Diseases

Diabetes, a disease whose incidence is rising continuously globally and is estimated to increase even more in the coming years [77], has been associated with an increased risk of developing GC [78]. Hyperglycemia, via the generated ROS, induces DNA damage, promoting EMT transition and leading to gastric mucosa damage. Insulin resistance contributes to GC development and progression via inflammation and activation of the nuclear factor-kB (NF-kB) pathway. Increased levels of insulin-like growth factor 1 (IGF-1) induced by hyperinsulinemia act as a pro-mitogen, which decreases apoptosis in tumor cells. Insulin receptors activate the phosphatidylinositol 3-kinase (PI3K) protein kinase B (AKT)/mammalian target of the rapamycin (mTOR) signaling pathway involved in GC proliferation and survival [79]. Patients with GC in early stages have increased levels of fasting glucose, fasting insulin, total cholesterol, and homeostasis model assessment of insulin resistance (HOMA-IR) compared with healthy patients [80]. HOMA-IR represents the fasting plasma insulin (mU/L) × fasting plasma glucose (mmoles/L)/22.5 and is a test for insulin resistance [80]. Furthermore, consumption of whole grains protects against GC, while ingestion of refined cereals can be a risk factor for GC [81], because the processing grains removes their fiber and micronutrient content [82].

Bile reflux, age, and GC family history are independent risk factors for the development of precancerous gastric lesions and GC [71,83]. Gastroesophageal reflux disease represents another contributing factor in populations with elevated incidence of cardia GC [84]. Acid reflux is characterized by the release of acids and enzymes, disturbing the esophageal barrier and leading to esophageal tissue and mucosa damage and inflammation [85].

### 2.4. Genetic Factors

Regarding hereditary GC cases, they account for 1–3% of all GC patients and are divided in hereditary diffuse gastric cancer (HDGC), gastric adenocarcinoma, proximal polyposis of the stomach (GAPPS), and familial intestinal gastric cancer (FIGC) [14]. HDGC is an autosomal dominant cancer syndrome, induced by the inactivation of germline mutations in the *CDH1* tumor suppressor gene and minor mutations in *CTNNA1*, especially for family cases [86].

Lynch syndrome represents one of the most common hereditary gastrointestinal cancer syndromes, with an extremely poor prognosis and a frequency of 1:279 [87]. Lynch syndrome is characterized by several gene mutations of DNA mismatch repair, including *MSH2*, *MLH1*, *MSH6*, and *PMS2*. *MSH2* mutation was observed in GC patients who developed the intestinal type (62–79%), and 23–32% of cases were diffuse or poorly differentiated [88].

Van Beek et al. detected that the most frequently mutated genes in GC are *ARID1A*, *CDH1*, *ERBB3, KRAS, PIK3CA*, and *TP53*. Among them, the *TP53* mutation is found in 50% of GC cases. The incidence of *TP53* mutations is influenced by race. African American patients present 89% *TP53*-related incidence, Asian patients 56%, Hispanic patients 43%, and Caucasian patients 40% *TP53*-related incidence [89]. Moreover, *TP53* loss and AKT overexpression are also correlated with GC development [90]. The genetic aberration will induce PI3K/AKT/mTOR overactivation, leading to anti-apoptosis and pro-survival properties [91]. An important characteristic of tumors, including gastric carcinogenesis, involves alterations and dysregulation of the extracellular matrix (ECM). Therefore, several ECM components, such as MMPs and collagen, play key roles in GC pathogenesis, promoting tumor migration and metastasis [92].

## 3. Overview of the PI3K/AKT/mTOR Signaling Pathway

This molecular pathway has two important components: PI3K and its downstream molecule, called protein kinase B, a serine/threonine kinase (PKB or AKT). Activation of this pathway starts with binding by the receptor tyrosine kinase (RTK) or G protein-coupled receptors (GPCRs) of various compounds [93]. The name PI3K gathers three distinguished classes of related kinases, by function and structure, involved in the phosphorylation of the inositol ring 3′-OH group from phospholipids [94]. Class IA PI3K is a heterodimer composed of a regulatory subunit (p85α/p55α/p50α, p85β, or p55γ, encoded by *PIK3R1*, *PIK3R2*, and *PIK3R3*, respectively) [95] and a catalytic subunit (p110, p110α, p110β, or p110δ, which are encoded by *PIK3CA*, *PIK3CB*, and *PIK3CD*) [95,96]. Moreover, the *PIK3CA* gene that encodes the catalytic subunit p110α is one of the frequently mutated oncogenes in various human cancers [97].

The second class, IBPI3K, is also a heterodimer consisting of one regulatory subunit, p101, and the catalytic subunit, p110γ. Class II PI3K has a single catalytic subunit, PI3KC2α, PI3KC2β, and PI3KC2γ, that can be activated by cytokine receptors, RTKs, and also integrins. The last one, class III PI3K, has only a catalytic subunit, VPS34 [96]. Regarding distribution, PI3Kα, PI3Kβ, PI3KC2α, and PI3KC2β isoforms are relatively ubiquitously expressed, PI3Kγ and PI3Kδ are found especially in leukocytes, while PI3KC2γ is found in the liver [98].

Various molecules, such as insulin, growth factors, cytokines, and hormones, bind to RTK or GPCRs and induce PI3K activation [99]. Further, PI3K is attached to the plasma membrane and catalyzes the phosphorylation of phosphatidylinositol (4,5)-bisphosphate (PIP2) to phosphatidylinositol (3,4,5)-trisphosphate (PIP3) [100]. PI3K/AKT/mTOR can be activated by overexpression of epidermal growth factor receptor (EGFR), vascular endothelial growth factor receptor (VEGFR), fibroblast growth factor receptor (FGFR), hepatocyte growth factor receptor (HGF), insulin receptor, hepatocyte growth factor (HGF), and leukocyte receptor tyrosine kinase [101].

Further, PIP3 is recruited to the plasma membrane as well as other kinases such as phosphoinositide-dependent kinase 1 (PDK1), which activate AKT by phosphorylation [102] in Thr 308 residue. In the phosphorylated form, p-AKT, activates other targets including mTOR, promoting protein synthesis, cell growth, survival, and motility [103]. AKT has three isoforms, AKT1, AKT2, and AKT 3. All the isoforms contain an N-terminal pleckstrin homology (PH) domain, a T-loop region with catalytic activity where Thr 308 suffers phosphorylation, and a C-terminal regulatory tail where Ser473 is also phosphorylated [104]. It is considered that AKT activates by phosphorylation more than 100 downstream targets, including glycogen synthase kinase3 (GSK-3), the Forkhead Box O (FoxO) proteins, and the mammalian target of rapamycin (mTOR) (Figure 1) [105]. AKT has three different subtypes, including AKT1/PKBa, AKT2/PKBb, and AKT3/PKBg, which are encoded by different genes [106].

The PI3K/AKT/mTOR signaling pathway plays a key role in cell survival via apoptosis-related gene inhibition, such as for Bcl-2-associated agonist of cell death (BAD), Bcl-2 associated X (BAX), caspase-9, GSK-3, and FoxO1 (Figure 2). This family of kinases also activates anti-apoptotic proteins, including NF-kB and cAMP response element-binding protein (CREB). mTOR has two distinct complexes, mTORC1 and the mammalian target of rapamycin complex 2 (mTORC2). Several molecules, including those associated with insulin, growth factors, rapamycin, certain amino acids, phosphatidic acid, and oxidative stress, affect mTORC1 activity [107].

Once activated, this serine/threonine protein kinase is involved in growth and survival regulation [108]. One of the most important downstream components of PI3K/AKT/mTOR is GSK-3. In mammals, GSK-3 has two forms, GSK-3α and GSK-3β, encoded by separate genes, *GSK3A* and *GSK3B* [109]. mTOR activation by AKT induces suppression of tuberous sclerosis complex 1 and 2 (TSC1/TSC2) [110]. Moreover, mTOR activation induces the activation of other downstream effectors, such as eukaryotic initiation factor binding protein 1 (4E-BP1) and p70S6 kinase (S6K) [111].

Activated mTORC1, via 4E-BP1 and S6K1, regulates important cellular processes, including mRNA transcription, cell autophagy, and protein synthesis [112]. In addition, mTORC1 induces the phosphorylation of ribosomal protein S6 kinase (pS6K) and 4E-BP1, which are involved in protein biosynthesis, ribosome biogenesis, and cell growth [113]. Via 4E-BP1 and S6K, mTOR controls cell growth and metabolism and also cell cycle progression [114].

mTORC1 is formed by Raptor and PRAS40, while mTORC2 contains Rictor and mSim1 [115]. mTORC2 regulates cell metabolism and survival and also the cytoskeleton [116]. PI3K, *Ras*, AMP-activated protein kinase (AMPK), TSC1/2, and S6K regulate mTORC2 activity. Therefore, S6K inhibits mTORC 2 activity by negative feedback regulation of the PI3K/AKT pathway [117]. Phosphatase and tensin homolog (PTEN) represent the negative regulator of PI3K/AKT/mTOR, involved in dephosphorylation of PIP3 to PIP2. Loss of *PTEN* is frequently met in cancer, leading to PI3K/AKT/mTOR hyperactivation [118,119]. The *PTEN* gene is located on chromosome 10q23.3, and its inactivation is correlated with GC initiation and development. Therefore, PTEN could be considered a prognostic biomarker for GC [120]. RTK and GPCR activation by various target molecules induces further mitogen-activated protein kinase (MAPK) complex activation [121].

## 4. Overview of the MAPK Signaling Pathway

MAPK cascades contain three main kinases, MAPK, MAPK kinase (MAPKK), and MAPKK kinase (MAPKKK) [122], which are involved in the phosphorylation of numerous downstream proteins. This molecular pathway regulates different cellular processes, such as proliferation, differentiation, stress response, and apoptosis [123]. MAPKs are serine/threonine protein kinases that are highly conserved in eukaryotic organisms. These proteins are encoded by different genes that are specific for MAPK (MPK), MAPKK (MKK or MEK), and MAPKKK (MEKK) [124]. MAPK has three major subfamilies: the first includes the extracellular-signal-regulated kinases (ERK MAPK); the second includes the c-jun N-terminal kinase or stress-activated protein kinases (JNK or SAPK); and the last includes MAPK14 [125]. Moreover, ERK 1 and 2 are known as MAPK3 and MAPK1, the atypical ERK3 and 4 are known as MAPK6 and MAPK4, and the stress-activated MAPKs p38α and β are known as MAPK14 and MAPK11 [126]. ROS can regulate ERK activation, promoting tumor signal transduction and further enhancing cell proliferation and survival. The MAPK/ERK pathway is formed by three MAPKKK types, including A-Raf, B-Raf, and Raf-1 or C-Raf kinases [127].

The MAPK-big signaling pathway, also known as Ras/Raf/MEK/ERK, is involved in inter-and intra-cellular communication. These kinases also regulate cell growth, survival, differentiation [128], and stress responses [129]. The central MAPK kinase, called MKK or MEK, is found in mammalian cells (MEK1/2) and regulates growth and development. The MKK3/6→p38 pathway regulates inflammation, while MKK4/7→JNK regulates apoptosis and processes-associated disease [130]. In addition, it was observed that the MAPK14/p38 kinase family is affected by environmental stress. Activation of p38 increased the risk of carcinogenesis and metastasis in various human malignancies [131]. The MAPK cascade can be activated by various molecules, including those associated with growth factors, immune system mediators [130], cytokines, hormones, oxidative stress, and endoplasmic reticulum stress [132].

Furthermore, *H. pylori* has the ability to activate PI3K/AKT/mTOR and MAPK pathways [88]. The Ras-MAPK ERK1/2 activate hundreds of transcriptional factors. Therefore, with the activation of cyclin D, Ras-MAPK regulates the progression of cells from G1 to the S phase of the cell cycle [133]. In normal cells, the activation of this signaling pathway starts with the binding of mitogens, hormones, or other stimulants to the tyrosine kinase receptors, which trigger dimerization and activation of oncogenic Ras from Ras-GDP, the inactive form, to the active one, Ras-GTP [134]. Moreover, growth factor receptor-bound protein 2 (GRB2) suffers phosphorylation in the Src homology domain (SH2) and activates Ras-GTP via the SOS system. Active Ras induces the formation and activation of the MAPK complex with its downstream kinases Raf, MEK1/2, ERK1/2, and other types of proteins. Moreover, the MAPK cascade may potentiate the activity of the PI3K/AKT/mTOR pathway [135].

An important member of the small GTPase Ras family is Rap1b. Ras is involved in the transformation of the inactive form, guanosine diphosphate (GDP) and the active form, 5′-adenosine triphosphate (GTP), regulating cell growth and differentiation. Rap1 induces ERK/MAPK activation via B-Raf, being implicated in cell–matrix adhesion and cell–cell junctions. Rap1b is overexpressed in various forms of cancer, including colorectal carcinoma, ovarian carcinoma, esophageal squamous carcinoma, squamous carcinoma, and GC. Higher Rap1b-expression is correlated with poor prognosis in GC. In vitro studies reported that Rap1b knockdown enhances GC apoptosis [136]. Dysregulation of two kinases from the MEKs family within the p38 pathway, MEK3 and MEK6, are associated with inflammatory diseases, various types of cancer, and neurological and cardiovascular disorders [137]. Among the first oncogenes that were discovered are *the Ras* gene, encoded by rat sarcoma viruses, including *v-Hras* and *v-Kras*, and representing the mutant forms of the cellular protooncogenes [138]. *Ras* mutation is the most frequent oncogene in human cancer [139]. The Raf-ERK1/2 and JNK cascades are the main regulators of malignant transformation, being involved in cell proliferation, migration, and survival [140].

## 5. PI3K/AKT/mTOR and MAPK Pathways in Gastric Cancer

PI3K/AKT/mTOR signaling pathway dysregulation is found in various pathologies, including cancer progression such as in GC. This pathway is involved in various cancer-vital processes, such as apoptosis, autophagy, cell growth, survival, and proliferation [141,142,143,144,145]. This signaling pathway is one of the main regulators of the cell cycle, growth, proliferation, metastasis, apoptosis, and autophagy [146]. In GC, PI3K/AKT/mTOR inhibits apoptosis and induces the chemo-resistance phenotype, metastasis, angiogenesis, and EMT [147] (Figure 2).

Besides the catalytic subunit of Class IA PI3K, other compounds such as the Class IA p85α-regulatory subunit, AKT, mTOR, and eukaryotic translation initiation factor 4E (eIF4E) possess oncogenic potential [148]. The downstream effectors of the PI3K/AKT/mTOR signaling pathway are usually dysregulated in the majority of solid tumors [149]. Moreover, PI3K/AKT/mTOR also controls cancer metabolism and genomic instability [150], having immunomodulatory potential as well [151].

In a sufficient supply of nutrients and energy, activated AKT will phosphorylate mTOR, further promoting tumor cell growth [152]. A very important component of this pathway, mTOR, is upregulated in malignancies, suppressing autophagy [153]. AKT was overexpressed in various neoplastic pathologies, such as pancreatic and ovarian cancer, as well as in gastric carcinoma, where the first isoform of AKT was found to be overexpressed [154]. GC-PI3K/AKT activation is induced by aberrant epigenetic regulations, necessary for GC development [155]. It is considered that *PIK3CA* is the second most common mutated gene in GC, responsible for 4–25% of cases, and also found in 85% of EBV GC cases [156]. Iranpour and his research group analyzed 100 patients with GC who underwent surgical resection and detected mutations in exon 20 of the *PI3KCA* gene in 11% of GC patients [157].

In gastric malignant tumors, p-AKT had elevated levels in 74% to 78%, being associated with GC angiogenesis and lymph nodes, leading to tumor invasion [120,158]. Overexpression of p-AKT is significantly correlated with HER2 overexpression and not with *PIK3CA* mutations. Moreover, the loss of *PTEN* is associated with GC initiation and development [120].

GC-EMT implies that the gastric cells lose their identity and acquire a mesenchymal phenotype with upregulation of N-cadherin, MMPs, and vimentin and downregulation of E-cadherin. PI3K/AKT/mTOR plays a pivotal role in EMT, leading to apoptosis resistance, invasion, and metastasis [159]. p-AKT induces dysregulation of the cell cycle, apoptosis suppression, and finally activation of angiogenesis. p-AKT will induce the phosphorylation of other proteins, including GSK-3β, BAD, caspase-9, and FoxOs [158].

Infection of gastric-epithelial cells with *H. pylori* will activate PI3K/AKT/mTOR and MAPK cascades, inducing the transformation of epithelial cells into neoplastic cells through changes including apoptosis, proliferation, and differentiation. The increased release of proinflammatory cytokines implies a higher production of free radicals, inducing DNA methylation [160].

RTK/MAPK pathway alteration, together with *TP53* mutations, was detected in GC subtypes. A cell motility gene, *CDH1*, is usually mutated in GC subtypes and induces the activation of the RTK/MAPK cascade [161]. The *PIK3CA* gene mutation rate was increased in GC, compared with *AKT1*, *AKT2*, and *AKT3*-mutations genes. Moreover, A*KT2* was significantly increased in EBV-positive GC as compared to EBV-negative GC. *AKT2* mutation was correlated with a poor survival rate in EBV-positive GC [162]. HER2 is frequently overexpressed in GC, leading to PI3K/AKT/mTOR and MAPK signaling pathway activation via RTK receptors and to increased cell proliferation, invasion, and metastasis [163].

*Raf-1* rs3729931, *HRAS* rs45604736, *MAPK1* rs2283792, and *MAPK1* rs9610417 genes are correlated with GC development [164]. Ras and Raf family member mutations induce the activation of RAF/MEK/ERK in GC [165]. GC-HER2 overexpression induces PI3K/AKT/mTOR and MAPK activation, promoting tumor-cell survival, proliferation, adhesion, and migration [166]. Lian et al. reported that IL-8 secretion via the ROS/NF-κB and ROS/MAPK (Erk1/2, p38)/AP-1 axis stimulates endothelial cell proliferation and angiogenesis in the GC tumor microenvironment [167]. Although ROS have a negative impact, they can also have positive roles, because they can be produced even by anti-cancer drugs to induce autophagy and apoptosis of malignant cells [168]. The PI3K/AKT/mTOR and Ras/Raf/MEK/ERK signaling pathways can be activated by common targets, such as Ras, and present some compensatory signaling properties; when one pathway is inhibited, the other is activated. Therefore, if mTOR is inhibited, PI3K will activate MAPK via Ras (Figure 2). These two cascades are also coactivated in various tumors, such as found in melanoma, prostate, and colorectal cancer [169]. Casein kinase II (CSNK2A1) is a serine/threonine kinase that is able to phosphorylate multiple substrates involved in cell cycle regulation, DNA repair and replication, transcription, apoptosis, and carcinogenesis. This kinase may be implicated in the processes of cancer invasion and migration via EMT transition and NF-kB signaling pathways. Regarding GC pathogenesis, CSNK2A1 is overexpressed, inducing the activation of AKT and mTOR, being involved in migration, invasion, and proliferation of GC cells [170].

## 6. PI3K/AKT/mTOR and MAPK Inhibitors

### 6.1. Inhibitors Used in Clinical Trials

In clinical trials, mTOR inhibitors (everolimus), dual PI3K/mTOR inhibitors, and ATP-competitive mTORC1/2 inhibitors are used for several types of tumors, including those associated with GC [171]. Prior to 2017, many drugs were tested in clinical trials for GC, with promising results coming from trastuzumab and ramucirumab [172] (Figure 2). Trastuzumab, a monoclonal antibody targeting HER2, was the first approved drug for GC therapy [173].

Hudis et al. conducted a phase I study to investigate the antitumoral activity of Trastuzumab (HER2 inhibitor) and MK-2206 (AKT 1 and AKT2 inhibitor) in patients with GC. The patients did not develop severe adverse reactions, with the most common side effects, reported in 40% of the patients, being fatigue, hyperglycemia, rash, nausea, decreased appetite, and liver cytolysis. The study reported that the combination of trastuzumab and MK-2206 had antitumoral activity in GC patients and was well tolerated [174].

A phase I study included patients with locally advanced and metastatic HER2-positive GC, breast, and gastroesophageal cancers that could not be removed by surgery and receiving a combination of MK-2206, lapatinib ditosylate, and trastuzumab. MK-2206 is an AKT inhibitor that, together with lapatinib, may inhibit tumor growth. It was expected that the administration of the monoclonal antibodies with MK-2206 and lapatinib would kill the cancer cells [175].

A phase II study used AKT inhibitor MK-2206 in GC/gastroesophageal junction cancers for patients who progressed after the first line of treatment. The study included patients with fasting serum glucose levels ≤ 150 mg/dL, adequate organ function, more than grade 2 malabsorption, or chronic diarrhea, who received the drug orally. MK-2206 did not increase antitumoral activity, with the overall survival (OS) being 5.1 months [176].

Everolimus, an oral mTOR inhibitor, has been used in preclinical, phase I, and phase II studies for GC patients. Therefore, Doi and his research team evaluated in a phase II study the antitumoral properties and safety of everolimus in pretreated advanced GC patients. The study observed a decrease in tumor size in 45% of the patients, with a median OS of 10.1 months and a median progression-free survival (PFS) of 2.7 months [177].

Everolimus was also administered in pretreated advanced GC patients enrolled in an international, double-blind, phase III study including six hundred fifty-six patients. The median OS was 5.4 months with everolimus treatment and 4.3 months with the placebo; the median PFS was 1.7 months with everolimus versus 1.4 months with the placebo. Therefore, the treatment with everolimus in advanced GC patients did not significantly improve the OS [178].

Shen et al. conducted a phase II trial where forty patients with chemotherapy-naive advanced GC received a low dose of everolimus in combination with cisplatin and 5-fluorouracil. Advanced GC patients received everolimus together with chemotherapy treatment (cisplatin, 5-fluorouracil, and leucovorin). The median PFS was only 6.9 months, and the OS was 10.5 months [179].

RAD001 or everolimus in combination with oxaliplatin and capecitabine was used in patients with advanced GC. Everolimus inhibits PI3K/AKT/mTOR and hypoxia-inducible factor 1-alpha (HIF 1α), an important factor that promotes angiogenesis and tumor growth. The patients received RAD001, capecitabine, and oxaliplatin in specific doses and split into five levels of treatment, with no results regarding OS and PFS [180]. The preclinical and clinical studies illustrated that oxaliplatin possesses antitumoral activity against gastrointestinal cancers without nephrotoxicity. Capecitabine has been approved by the Food and Drug Administration and the European Agency for the Evaluation of Medicinal Products (EMEA) for the treatment of patients with colorectal cancer. In addition, the combination of oxaliplatin and capecitabine was used in a multicenter phase II trial for patients with advanced hepatocellular carcinoma. The phase II trial study results were not very promising regarding PFS and OS [181]. The combination of two drugs, oxaliplatin and capecitabine, represents a platinum-based chemotherapy that initially was considered to have good effects on GC patients regarding survival, being able to suppress local recurrence. But patients started to develop oxaliplatin resistance, leading to chemotherapy failure [182]. Therefore, antiangiogenic drugs such as bevacizumab, sunitinib, and sorafenib have no efficacy for GC patients [183].

Lee and his research team conducted a single-center phase 2 trial of multiple solid malignant tumors harboring PI3K/AKT aberrations, where patients received TAS-117. TAS-117 is an AKT inhibitor with increased affinity for all three isoforms, which in vitro was able to inhibit breast, endometrial, lung, and ovarian cell culture line proliferation. This trial included thirteen patients divided into two groups, one with non-gastrointestinal (GI) cancer (breast, ovarian, endometrial, non-small cell lung cancer) and five patients with GI cancer (colon, rectal, gastric, and gallbladder), all having PI3K/AKT mutations. GI cancer patients received orally TAS-117 daily. Grade 3–4 related side effects for TAS-117 were anorexia and hyperglycemia. The study reported limited antitumoral activity for GI cancer patients, except for patients with ovarian and breast cancer [184].

A phase IB, multicenter, open-label dose escalation study tested the antitumoral effects of BYL719 (PI3K inhibitor) in combination with AUY922 (HSP90 inhibitor) in advanced or metastatic GC patients carrying the *PIK3CA* mutation or with HER2 amplification. The cycle of treatment lasted 28 days. The study did not report the effects of the treatment regarding PFS, OS, and adverse reactions [185].

Another Phase Ib/IIa, open-label, non-randomized, dose-escalation, multicenter study was conducted to evaluate GSK2636771 (PI3Kβ Selective Inhibitor) and paclitaxel, regarding safety and tolerability, for patients with PTEN-deficiency and advanced gastric adenocarcinoma. The patients received, separately, GSK2636771 and paclitaxel in different cycles of treatment and then the two drugs together [186].

PANGEA-IMBBP is a personalized clinical study where patients with gastro-esophageal adenocarcinoma received different treatments according to the MAPK/PI3K/AKT mutations identified. Therefore, patients with MAPK/PIK3K overexpression received standard cytotherapy and ramucirumab, and those with EGFR expression received standard cytotherapy and ABT-806. HER2 patients received standard cytotherapy plus trastuzumab. This clinical trial reported a PFS of 8.2 months. Also, various adverse effects were observed depending on the pathology and treatment, including gastrointestinal disorders, general disorders, infections, metabolic and nutrition disorders, musculoskeletal and connective tissue disorders, nervous system disorders, psychiatric disorders, and renal dysfunction [187], with cancer patients having a higher risk for acute kidney injury [188].

During 2014–2018, Lee and his research team conducted one of the most important clinical trials, called VIKTORY, which included seven hundred seventy-two patients with metastatic GC, divided into eight different groups according to the specific gene mutation. The patients received a specific inhibitor, such as capivasertib (AKT inhibitor), vistusertib (TORC inhibitor), or selumetinib (MEK inhibitor), as a second-line treatment. The treatment of metastatic GC patients with PIK3CA-capivasertib induces moderate antitumoral activity. The highest antitumoral activity in patients with *PIK3CA* mutations was reported after the treatment with capivasertib (also used as an AKT inhibitor)/paclitaxel. Regarding the other groups included in the study, the results were not very promising regarding the survival rate [189].

The randomized, double-blind, placebo-controlled, multicenter, phase II trial conducted by Bang and colleagues included 153 patients with locally advanced or metastatic GC or gastroesophageal junction cancer who were randomly assigned to receive ipatasertib (AKT inhibitor) or a placebo, plus mFOLFOX6 (1:1). The patients were split into various subgroups according to *PTEN* loss and PI3/AKT activation. The PFS was not statistically improved in patients with low PTEN and PI3K/AKT activation who received ipatasertib/mFOLFOX6 versus placebo/mFOLFOX6 [190].

A randomized, multicenter, open-label phase IIa trial included thirty patients with HER2-positive advanced GC or gastroesophageal junction cancer that received treatment divided into three weekly cycles. The patients received pertuzumab plus trastuzumab, which were well tolerated [191] (Table 1).

### 6.2. Inhibitors Used in In Vitro and In Vivo Studies

Treatment of GC cell lines with everolimus promotes decreased phosphorylation of S6K1 and 4E-BP1, induces G1 cell cycle arrest, and inhibits the proliferation of GC cells. In addition, the combination of everolimus with 5-fluorouracil, cisplatin, oxaliplatin, or docetaxel has the capacity to inhibit the growth of 5-flurouracil-resistant GC cell lines [192].

The well-known everolimus (RAD001) is an mTOR inhibitor widely used in GC treatment. The combination of everolimus with Rhein induces in GC-MGC803 cells a repression regarding viability, invasion, and proliferation. This combination affects the levels of E-cadherin, N-catherin, and vimentin by decreasing them. Also, everolimus-Rhein blocks the phosphorylation of PI3K/AKT/mTOR and suppresses cell proliferation marker Ki-67 [193]. Paclitaxel was frequently used in GC chemotherapy, but patients developed resistance. Activation of PI3K/AKT/mTOR and MAPK pathways may be involved in GC-paclitaxel resistance and may be correlated with apoptosis resistance. BEZ235, a dual PI3K/mTOR inhibitor showed promising results in vivo and in vitro against paclitaxel-GC resistance [194].

Using human gastric adenocarcinoma epithelial cell lines infected with *H. pylori*, Lee and his research team tested the effects of a carotenoid, astaxanthin, regarding P3K/AKT/mTOR and MMP expression. The study revealed that astaxanthin significantly reduces the expression of MMP-7 and 10 after 24 h of treatment. Moreover, this carotenoid had the ability to decrease the levels of PI3K/AKT/mTOR in GC cells infected with *H. pylori* [51]. Wortmannin is a Pan-class I inhibitor that exhibits PI3K-inhibitory properties but has limited clinical use due to biological instability and short half-life [195].

GSK1059615 is a dual inhibitor acting on PI3K and mTOR. Bei and colleagues used a cell line and treated human GC cells with GSK1059615 and observed that this inhibitor has the ability to block GC cell growth, proliferation, and survival, inducing apoptosis [196]. Another promising antitumoral drug against the PI3K/AKT/mTOR/S6K pathway is ABBV-744. This inhibitor was also used in GC cell line studies, where the results illustrated the inhibition of the PI3K cascade in a concentration-dependent manner [197].

Wang and colleagues used IGF-1, an AKT agonist, and LY294002, an inhibitor, on GC cell lines and observed a decreased expression regarding proliferation and angiogenesis [198]. Ipatasertib is a pan-AKT inhibitor with antitumoral activity, observed in several cancer cell lines and xenograft models. Regarding ipatasertib adverse events, the patients manifested diarrhea, rashes, nausea, and asthenia/fatigue. The patients did not have hyperglycemia after oral administration of ipatesertib [149].

Promising results regarding the fight against GC come from pectolinarigenin, which displays anti-inflammatory and antitumoral properties. Lee and his research team evaluated the anti-cancer properties of this compound, isolated from citrus fruits, using human gastric cell lines. The study revealed that this natural flavonoid had the capacity to induce G2/M phase cell arrest, autophagy, and apoptosis in GC cell lines through downregulation of the PI3K/AKT/mTOR cascade acting on p-EBP1, p-S6K, and p-eIF4E compared with untreated cells [199].

Another important compound extracted from the seed oil of Nigella sativa, thymoquinone, inhibits the development of several types of cancer, such as colorectal cancer, pancreatic cancer, prostate cancer, and breast adenocarcinoma. Moreover, in the GC cell line, this lactone decreases the levels of p-AKT (Ser473 and Thr 308), p-mTORC1, and p-mTORC2 in a dose-dependent manner [200]. It is well known that hypoxia and acidity modulate the TME, inducing cancer progression, invasiveness, metastasis, and tumor resistance. The acid TME is characterized by over-expression of H^+^ transporters. Therefore, proton pump inhibitors have been used both in vivo and in vitro to block the H^+^ transporters, inducing apoptosis. Moreover, the acidic medium may activate the PI3K/AKT/mTOR pathway, enhancing the phosphorylation of AKT. To test these observations, Chen and colleagues treated human gastric adenocarcinoma cell lines with a proton pump inhibitor (esomeprazole) in different concentrations and observed the inhibition of the PI3K/AKT/mTOR/HIF-1α cascade. The treatment of GC cells with rapamycin also showed that the intracellular expression of mTOR was significantly reduced [201]. Yao and his research team observed in vitro that the mTOR inhibitor rapamycin decreases GC cell proliferation and invasiveness [202].

Cytochrome P450 family 2 subfamily E polypeptide 1 (CYP2E1) dysregulation is involved in GC development and progression, promoting proliferation and invasion and inhibiting GC cell apoptosis. In addition, CYP2E1 overexpression activates mTOR and further PI3K/AKT cascade. Wang and colleagues investigated the effects of LY294002, a PI3K inhibitor, on GC cells, observing that this inhibitor downregulates the expressions of PI3K, AKT, mTOR, S6K, and CYP2E1 [203]. Moreover, an anti-helminthic used in veterinary medicine, rafoxanide, has anti-cancer properties, inhibiting the growth of colorectal, multiple myeloma, and skin cancers. Regarding the GC cell line, rafoxanide blocks its growth, inducing apoptosis and autophagy in cells. Rafoxanide inhibits PI3K and decreases GC tumor growth [204]. Yuan and colleagues investigated the effects of alisertib on NCI-N7 and AGS cells and reported that this compound had anti-apoptotic properties in both lines. Moreover, it inhibits PI3K/AKT/mTOR and p38-MAPK pathways and activates 5′-AMP-activated protein kinase, inducing a pro-autophagic effect on NCI-N7 and AGS cells [205].

Promising in vivo and in vitro results against GC come from NVP-BEZ235, a dual PI3K/mTOR inhibitor. Li and his research team, using human GC-MKN-45 cells and a nude mice xenograft with MKN-45 cells, evaluated the effect of NVP-BEZ235 and 5-fluorouracil alone or in combination, regarding proliferation, invasion, apoptosis, and chemoresistance. In vitro results suggested that the combination of NVP-BEZ235 and fluorouracil has good effects on MKN-45 cells, inhibiting proliferation, migration, and invasion. The in vivo results were also good, because NVP-BEZ235 with 5-fluorouracil inhibited growth and induced MKN-45 cell apoptosis. NVP-BEZ235 acts on the PI3K/AKT/mTOR cascade, reducing the levels of MMP-9 and VEGF and increasing the levels of proapoptotic compounds such as BAX [206].

CMG002, a PI3K/mTOR dual inhibitor, was combined with chloroquine (autophagy inhibitor) to induce apoptosis in EBV-GC cells. EBV and mock-infected AGS and NUGC-GC cells were treated with CMG002 with or without chloroquine. The combination of CMG002 and chloroquine showed increased therapeutic efficacy against EBV-GC [207].

The effects of sinulariolide were investigated by Wu and colleagues using GC cell lines AGS and NCI-N87 regarding antimigratory and anti-invasive properties. The cell line studies observed that increased concentrations of sinulariolide decreased AGS and NCI-N87 cell viability, migration, and metastasis. This compound was able to downregulate the expression of p-PI3K, p-AKT, p-mTOR, p-JNK, p-JUN, p-p38, and p-ERK in AGS and NCI-N87 cells. It is known that activation of ERK by phosphorylation promotes cancer cell proliferation, invasion, and movement. Therefore, MAPK and PI3K compounds’ inhibition by sinulariolide decreases GC invasion and metastasis. Sinulariolide has the capacity to block EMT [208]. Yang and his research team investigated the antitumoral properties of deltonin, the active compound from *Dioscorea zingiberensis*, in GC cell lines. Deltonin was able to enhance GC cell line apoptosis and DNA repair in a dose-dependent manner. Moreover, deltonin was able to decrease the phosphorylation rate of PI3K, AKT, mTOR, and p38-MAPK. Deltonin also induced the chemosensitivity of GC cells to cisplatin via enhanced apoptosis and decreasing GC proliferation and growth [209]. Fukuoka et al. tested the effects of DIACC3 and the S6K/AKT 1/3 dual inhibitor in vivo and in vitro using preclinical models. In addition, researchers evaluated the efficacy of DIACC3010 in combination with trastuzumab in HER2-positive cell lines and models, where 21 HER2-negative PDX models reported that DIACC3010 inhibited growth by 38%. The study reported that cell lines that were sensitive to DIACC3010 had *PIK3CA* mutation and decreased p-ERK. One-third of all GC PDX models that used DIACC3010 monotherapy reported significant efficiency. ERBB2-amplified cell lines and PDX models did not respond to DIACC3010 monotherapy but were sensitive to trastuzumab treatment alone or in combination with DIACC3010 [210].

Zhang et al. tested an active compound from plants, calycosin, in GC cell lines and observed that it was able to induce apoptosis via ROS-mediated MAPK/signal transducer and activator of transcription 3 (STAT3)/NF-kB signaling pathways. Therefore, calycosin seems to be able to downregulate the anti-apoptotic proteins (Bcl-2) and increase the expression of pro-apoptotic proteins (caspase-3, cytochrome C) via the PI3K/AKT/mTOR pathway. In AGS cells, calycosin regulates MAPK/STAT3/NF-kB cascade proteins by binding to DNA or transcriptional factors, inducing apoptosis of these cells [211].

Another promising study investigated the effects of glycitein on human gastric cells. Glycitein had the capacity to decrease mitochondrial transmembrane potential, increasing mitochondrial-related apoptosis and promoting G0/G1 cell cycle arrest. Moreover, this isoflavone was able to activate MAPK and inhibit STAT3 and NF-kB [212].

Although in vitro studies showed that trastuzumab presents antitumoral activity in HER2-positive breast cancer and GC, an elevated number of patients did not respond to trastuzumab. Therefore, the combination of trastuzumab and osthole, an active coumarin, possesses antitumoral activity in HER2-overexpressing tumors. Yang Y. and colleagues tested the antitumoral activity of the two compounds on the N87-GC cell line and SK-BR- breast cell line, both HER2-overexpressing cell lines. The treatment of cell lines only with osthole is conducive to cell cycle arrest on the G2/M phase and apoptosis. The treatment with trastuzumab plus osthole was more potent, inducing and enhancing apoptosis. The study reported that the combination of these two drugs significantly decreases the AKT and MAPK dephosphorylation in N87-GC cells. Using N87 tumor xenografts, the treatment with trastuzumab and osthole had synergetic effects, especially on the AKT signaling pathway [213].

Zhang and his research team evaluated the antitumoral activity of dihydroartemisinin on human GC cell line BGC-823. Dihydroartemisinin inhibits cell viability of the GC cell line in a dose- and time-dependent manner by upregulation of BAX and cleaved caspase-3 and 9, and downregulation of the Bcl-2/BAX ratio. In BGC-823 xenograft models, dihydroartemisinin induces apoptosis via JNK1/2 and p38 MAPK signaling pathways [214].

Overexpression of PI3K induces trastuzumab resistance in HER2-positive GC. Therefore, Mezynski and colleagues, using several gastric cell lines, tested the anti-cancer properties of copanlisib (PI3K inhibitor) in combination with refametinib (MEK1/2 inhibitor) and an HER2 inhibitor. The combination of copanlisib and trastuzumab significantly decreases the growth in several GC cell lines. Moreover, copanlisib together with refametinib and trastuzumab showed promising results for HER2-positive GC patients [30]. Using several cell lines, Sheng et al. evaluated the anti-cancer properties of zeaxanthin and reported that this common carotenoid was able to induce apoptosis in a gastric cell line by increasing the levels of pro-apoptotic compounds. Furthermore, zeaxanthin induces the formation of ROS, increases the levels of p-JNK and p-p38, and decreases the levels of p-ERK, p-AKT, and NF-kB. The activation of kinases JNK and p38 MAPK induces apoptosis, while phosphorylated ERK induces cell survival. NF-kB is implicated in cell growth, survival, adhesion, inflammation, and differentiation. Therefore, decreased levels of NF-kB observed in gastric cell lines illustrated the antitumoral activity of zeaxanthin. Furthermore, zeaxanthin-ROS overproduction represents a mechanism to kill GC cells [215].

Capasso et al. tested in vitro the effects of several inhibitors, such as erlotinib (EGFR inhibitor), the MEK inhibitor (BAY-86-9766), trastuzumab (HER2 inhibitor), and GDC-0980 (PIK3K/mTOR inhibitor) alone or in combination on various types of GC cells. These cells have an overexpression of EGFR, HER2, MAPK, AKT, and 4E-BP1. The treatment of the HER2-GC cell line with trastuzumab induces growth inhibition only in one cell line. The combination of MEK and PIK3CA/mTOR inhibitors in various concentrations inhibits the growth of several GC cells by reducing the level of AKT, p-AKT, MAP, p-MAP, 4E-BP1, and p-4EBP1 [216]. Another promising drug candidate in the fight against GC may be ABBV-744, which was evaluated in GC cell lines and possesses antitumoral activity by promoting mitochondria damage, ROS overproduction, cell cycle arrest, and apoptosis in cancer cells. Moreover, this compound had the capacity to induce autophagy in GC cells by the inactivation of PI3K/AKT/mTOR and MAPK cascades. Studies performed in vivo using a mouse xenograft sustain the results from cell culture, promoting GC cell growth inhibition via autophagy [197].

## 7. Conclusions

GC remains a serious health problem, especially in Eastern Asian countries and the southern region of India. Endoscopy represents the gold standard method, used in some countries as a mass screening program, which significantly improves the survival rate, especially for GC patients at stages I and II. Crowded families with low socioeconomic status have an elevated risk of *H. pylori* infection, which can develop, in particular, into the non-cardia subtype. Diet plays a key role in GC development, because some eating habits may irritate the stomach mucosa. Genetic mutations induce dysregulation of ECM, such as MMPs promoting tumor migration and metastasis.

In various malignancies, including GC, PI3K/AKT/mTOR and MAPK signaling pathways are overexpressed, being involved in processes such as growth, survival, autophagy, and proliferation. The PI3K/AKT/mTOR pathway inhibits apoptosis, promoting chemo-resistance, metastasis, angiogenesis, and EMT.

Although PI3K/AKT/mTOR and MAPK signaling pathway inhibitors have shown promising results in cell line studies, in vivo studies and clinical trials have not shown as encouraging results, with OS and PFS not significantly improved. In addition, most inhibitors showed adverse effects. Dual inhibitors that simultaneously target both signaling pathways have shown better results in cell line and animal studies using mouse xenografts but are under-investigated in clinical trials. So far, the results obtained in clinical trials have not been promising, and multiple adverse effects have been registered. However, given the in vitro and in vivo results, we consider it necessary to test more dual inhibitors in clinical trials and to develop new ones with reduced adverse effects.

## Figures and Tables

**Figure 1 ijms-25-01848-f001:**
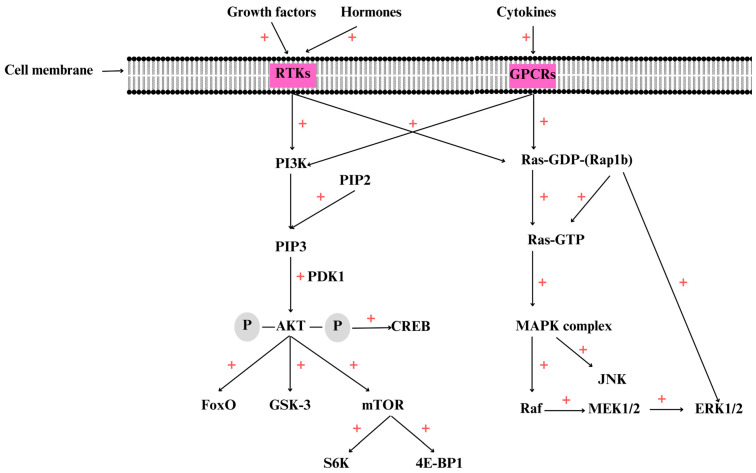
Activation of PI3K/AKT/mTOR and MAPK signaling pathways. The binding of various growth factors, hormones, and cytokines to receptor tyrosine kinase (RTK) or G-protein-coupled receptors (GPCRs) leads to the activation of phosphatidylinositol 3-kinase (PI3K), which induces the phosphorylation of phosphatidylinositol (4,5)-bisphosphate (PIP2) to phosphatidylinositol (3,4,5)-triphosphate (PIP3), inducing protein kinase B (AKT) activation by phosphorylation by phosphoinositide-dependent kinase 1 (PDK1). Once activated, p-AKT activates other substrates by phosphorylation, such as the mammalian target of rapamycin (mTOR), glycogen synthase kinase-3 (GSK-3), the Forkhead box proteins (FoxOs) and c Adenosine monophosphate (cAMP) response binding protein (CREB). mTOR activates other proteins such as eukaryotic initiation factor binding protein 1 (4E-BP1) and ribosomal protein S6 kinase beta-1 (S6K), also known as p70S6 kinase. Mitogen-activated protein kinase (MAPK) complex activation starts with the formation of Ras-GTP from the inactive form Ras-GDP, which further activates the downstream substrates, including Raf, MAPK kinase (MAPKK or MEK), extracellular-signal-regulated kinases (ERK1/1), and c-jun N-terminal kinase or stress-activated protein kinase (JNK). In healthy cells, activation of these signaling pathways sustain the normal functions of the cells. Note: **+** denotes activation.

**Figure 2 ijms-25-01848-f002:**
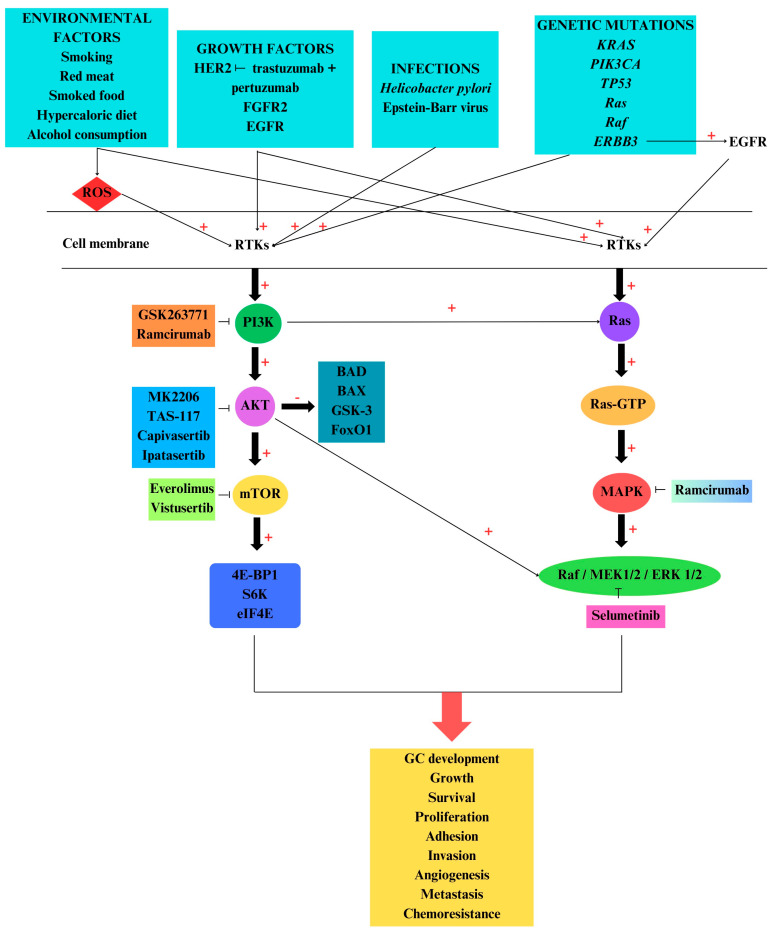
Phosphatidylinositol 3-kinase (PI3K) protein kinase B (AKT)/mammalian target of rapamycin (mTOR), PI3K/AKT/mTOR, and mitogen-activated protein kinase (MAPK) signaling pathway activation in gastric cancer (GC). Environmental factors, growth factors, genetic mutations, and infections with *Helicobacter pylori* and Epstein–Barr virus (EBV) activate both PI3K/AKT/mTOR and MAPK signaling pathways via receptor tyrosine kinases (RTKs). Further, PI3K is activated by phosphorylation, inducing the activation of AKT and downstream activation of other protein kinases such as the mammalian target of rapamycin (mTOR). In a similar manner, MAPK cascade is activated, and together with PI3K/AKT/mTOR, contributes to GC development, sustaining growth, survival, proliferation, angiogenesis, and metastasis. In clinical studies, several inhibitors of these protein kinases have been used. Therefore, PI3K, AKT, mTOR, MAPK kinase (MAPKK or MEK1/2), and human epithelial growth receptor 2 (HER2), showed different responses regarding the survival rate. Note: **+** denotes activation; **−** denotes inhibition.

**Table 1 ijms-25-01848-t001:** The main PI3K/AKT/mTOR and MAPK inhibitors used in clinical trials.

Trial Type	Drug	Target	Reference
Phase I	Trastuzumab	HER2	[174]
	MK-2206	AKT 1 and AKT 2	
Phase I	MK-2206+	AKT	[175]
	lapatinib ditosylate + trastuzumab	HER2	
Phase 2	MK-2206	AKT	[176]
Phase II	Everolimus (RAD 0001)	mTOR	[177]
International, double-blind, phase III	Everolimus (RAD001)	mTOR	[178]
Phase II	Everolimus + cisplatin + 5-fluorouracil Everolimus + cisplatin, 5-fluorouracil + leucovorin (advanced GC)	mTOR	[179]
Single-center, phase 2	TAS-117	AKT	[184]
Phase IB, multicenter	BYL719	PI3K	[185]
Phase Ib/IIa, open-label, non-randomized	GSK2636771 + paclitaxel	PI3Kβ	[186]
First pilot metastatic	Cytotherapy + ramucirumab	PI3K and MAPK	[187]
	ABT-806	EGFR	
	standard cytotherapy + trastuzumab	HER 2	
VIKTORY	capivasertib	AKT, PI3K	[189]
	vistusertib	TORC	
	selumetinib	MEK	
	capivasertib + paclitaxel	PI3K/AKT	
The randomized, double-blind, placebo-controlled, multicenter, phase II trial	Ipatasertib + mFOLFOX6	AKT	[190]
A randomized, multicenter, open-label phase IIa	Pertuzumab + trastuzumab	HER2	[191]

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
