# Peer review of "Targeting PI3K/AKT/mTOR and MAPK Signaling Pathways in Gastric Cancer"

_ijms, 2024, doi:10.3390/ijms25031848_

Round 1

Reviewer 1 Report

Comments and Suggestions for Authors

ijms-2811686

Type of manuscript: Review

Title: Targeting PI3K/AKT mTOR and MAPK signaling pathways in Gastric Cancer

Authors: Diana-Theodora Morgos, Constantin Stefani *, Daniela Miricescu *, Maria Greabu, Silviu Stanciu, Silvia Nica, Iulia-Ioana Stanescu-Spinu, Daniela Balan, Andra-Elena Balcangiu-Stroescu, Elena Claudia Coculescu, Dragos Eugen Georgescu, Remus Nica

This paper is a review article on the PI3K/AKT/mTOR-MAPK signaling pathway in the development of gastric cancer. Overall, it is well-written; however, it requires more detailed corrections and updates.

[Major concerns]

1.    The Mediterranean diet at Line 161: The statement suggests a potential positive association between following the Mediterranean diet and a reduced risk of gastric cancer. The Mediterranean diet is generally characterized by a high intake of fruits, vegetables, whole grains, olive oil, and fish, with moderate consumption of red wine. These dietary components are believed to contribute to overall health and may have protective effects against certain diseases, including cancer. However, it's important to note that individual factors, lifestyle choices, and genetic predispositions also play significant roles in cancer development. While there is evidence supporting the health benefits of the Mediterranean diet, further research and clinical studies are needed to establish a clear cause-and-effect relationship between this diet and a decreased incidence of gastric cancer. Therefore, it might be advisable to express it with a bit more nuance, contrary to what is stated on Line 161.

2.    Abbreviations: The use of abbreviations when writing a paper has many advantages besides simplicity of expression. To use an abbreviation, first write the abbreviation in parentheses after the full name, and then use the abbreviation from Introduction to the final Conclusion. Only in Abstract and Figure legend do it separately. If an abbreviation is not used more than twice, there is no need to define it, so please delete it.

3.    English: The English composition of the paper is generally well-done. However, some names of disease or compound names are written in uppercase letters even though they are not the first letter of the sentence or proper nouns. Please make corrections throughout the text and in the figures. Additionally, the notation of certain words is inconsistent. English proofreading by an expert with experience in this field is required. Examples: CagA at Line 61 vs. Cag A at Line 62; Cardia at Line 74; Bcl2 and Bcl-2 at Lines 687 and 709, etc.

4.    Gastrointestinal Tumors at Line 1120: While discussing cancer in all other organs, is it necessary to have a subheading specifically labeled 'Gastrointestinal Tumor' here? The content described pertains to cancer, and it is well understood by the authors, as well as commonly acknowledged, that tumor and cancer are as different as day and night.

5.    Notation: The notation for proteins and genes is different. For genes, it should be indicated in italics. Examples: TP53 at Line 88, etc.

6.    In the main text, referencing a bibliography can be done by mentioning the first author's last name only. Examples: Yuan P at Line 134; Li Y at Line 181, etc.

7.    H. pylori vs. H.pylori: I was quite surprised to see the use of these two notations. Please correct them properly.

[Minor concerns]

1.    Lines 143 and 144: Define HDL-C and LDL-C.

2.    Line 150: Define EMT.

3.    Line 157: Define HOMA-IR and explain it here.

4.    Line 161: ‘Mediterranian’ should be written as Mediterranean.

5.    Line 203: p53 is a protein, not a gene.

6.    Line 273: P70S6 should be written as p70S6.

7.    Figure 1: Only the title of Figure 1 is present, and there is a lack of explanation regarding the content of the figure. Additionally, list the abbreviations used in the figure in the Figure Legend, along with their respective full names. When there are multiple abbreviations, arrange them alphabetically.

8.    Line 670: Dioscorea zingiberensis should be written in italics.

9.    Line 687: Caspase-3 should be written as caspase-3.

10.    Lines 687 and 709: Bcl2 vs. Bcl-2.

11.    Lines 698, 700, and 757: HER-2 vs. -HER2 vs. HER 2.

12.    Line 702: ostiole? What is it?

13.    Lines 730 and 733: Define PIK3CA and explain about this.

14.    Line 755: ‘angiogeensis’ should be written as ‘angiogenesis’.

Overall, the manuscript can be considered to publication after major revision as indicated above.

Comments on the Quality of English Language

ijms-2811686

Type of manuscript: Review

Title: Targeting PI3K/AKT mTOR and MAPK signaling pathways in Gastric Cancer

Authors: Diana-Theodora Morgos, Constantin Stefani *, Daniela Miricescu *, Maria Greabu, Silviu Stanciu, Silvia Nica, Iulia-Ioana Stanescu-Spinu, Daniela Balan, Andra-Elena Balcangiu-Stroescu, Elena Claudia Coculescu, Dragos Eugen Georgescu, Remus Nica

This paper is a review article on the PI3K/AKT/mTOR-MAPK signaling pathway in the development of gastric cancer. Overall, it is well-written; however, it requires more detailed corrections and updates.

[Major concerns]

1.    The Mediterranean diet at Line 161: The statement suggests a potential positive association between following the Mediterranean diet and a reduced risk of gastric cancer. The Mediterranean diet is generally characterized by a high intake of fruits, vegetables, whole grains, olive oil, and fish, with moderate consumption of red wine. These dietary components are believed to contribute to overall health and may have protective effects against certain diseases, including cancer. However, it's important to note that individual factors, lifestyle choices, and genetic predispositions also play significant roles in cancer development. While there is evidence supporting the health benefits of the Mediterranean diet, further research and clinical studies are needed to establish a clear cause-and-effect relationship between this diet and a decreased incidence of gastric cancer. Therefore, it might be advisable to express it with a bit more nuance, contrary to what is stated on Line 161.

2.    Abbreviations: The use of abbreviations when writing a paper has many advantages besides simplicity of expression. To use an abbreviation, first write the abbreviation in parentheses after the full name, and then use the abbreviation from Introduction to the final Conclusion. Only in Abstract and Figure legend do it separately. If an abbreviation is not used more than twice, there is no need to define it, so please delete it.

3.    English: The English composition of the paper is generally well-done. However, some names of disease or compound names are written in uppercase letters even though they are not the first letter of the sentence or proper nouns. Please make corrections throughout the text and in the figures. Additionally, the notation of certain words is inconsistent. English proofreading by an expert with experience in this field is required. Examples: CagA at Line 61 vs. Cag A at Line 62; Cardia at Line 74; Bcl2 and Bcl-2 at Lines 687 and 709, etc.

4.    Gastrointestinal Tumors at Line 1120: While discussing cancer in all other organs, is it necessary to have a subheading specifically labeled 'Gastrointestinal Tumor' here? The content described pertains to cancer, and it is well understood by the authors, as well as commonly acknowledged, that tumor and cancer are as different as day and night.

5.    Notation: The notation for proteins and genes is different. For genes, it should be indicated in italics. Examples: TP53 at Line 88, etc.

6.    In the main text, referencing a bibliography can be done by mentioning the first author's last name only. Examples: Yuan P at Line 134; Li Y at Line 181, etc.

7.    H. pylori vs. H.pylori: I was quite surprised to see the use of these two notations. Please correct them properly.

[Minor concerns]

1.    Lines 143 and 144: Define HDL-C and LDL-C.

2.    Line 150: Define EMT.

3.    Line 157: Define HOMA-IR and explain it here.

4.    Line 161: ‘Mediterranian’ should be written as Mediterranean.

5.    Line 203: p53 is a protein, not a gene.

6.    Line 273: P70S6 should be written as p70S6.

7.    Figure 1: Only the title of Figure 1 is present, and there is a lack of explanation regarding the content of the figure. Additionally, list the abbreviations used in the figure in the Figure Legend, along with their respective full names. When there are multiple abbreviations, arrange them alphabetically.

8.    Line 670: Dioscorea zingiberensis should be written in italics.

9.    Line 687: Caspase-3 should be written as caspase-3.

10.    Lines 687 and 709: Bcl2 vs. Bcl-2.

11.    Lines 698, 700, and 757: HER-2 vs. -HER2 vs. HER 2.

12.    Line 702: ostiole? What is it?

13.    Lines 730 and 733: Define PIK3CA and explain about this.

14.    Line 755: ‘angiogeensis’ should be written as ‘angiogenesis’.

Overall, the manuscript can be considered to publication after major revision as indicated above.

Author Response

Dear reviewer,

On behalf of our research group, we would like to thank you for your time and your comments. We highly appreciated your recommendations and taking into consideration your suggestions, we made the following changes to the manuscript:

Major concerns

  1. The phrase regarding the Mediterranean diet has been rephrased to be more nuanced, lines 190-193.

  1. All the abbreviations from the article were checked and inserted where they were missing, lines: 209, 210; 231, 246, 247, 317, 371, 380, 495, 534, 723, 771, 772, 849, 850, 874.

​

  1. We corrected the mistakes from the article regarding cagA and vac A; Bcl-2, lines: 66, 340, 341, 358.
  2. Tumor versus cancer, we correct this confusion, lines: 53, 631, 477, 792.
  3. Genes were written in italics, lines: 33, 99, 100, 261, 263, 289, 359, 480.
  4. We wrote the author’s only the last name into the article, lines: 161, 212, 560, 606, 630, 692, 738, 739, 746, 762, 778, 789, 795, 811, 825, 833, 848, 878, 883, 893.
  5. pylori was corrected, lines: 109, 123, 124.

Minor concerns

  1. HDL-C and LDL-C were defined, lines: 171, 172.
  2. EMT was defined, line 98.
  3. HOMA-IR was defined and explained, lines: 230-233.
  4. „mediterranean „ was corrected, line 190.
  5. p53 was corrected, line 263.
  6. P70S6 was corrected, lines 362, 372, 750, 766, 791, 839.
  7. Explication and the full name for all the abbreviations from the figures 1 and 2 were included, figure 1 lines: 326-337; figure 2, lines: 444-455.
  8. Dioscorea zingiberensis was written in italics, line 834.
  9. caspase-3 was corrected, line 852.
  10. Bcl2 was corrected, lines: 340, 874.
  11. HER2 was corrected, lines: 90, 694, 696
  12. The word ostiole is wrong, the correct form is osthole, we corrected and explained into the article, lines 693-701.
  13. PIK3CA is the gene that encodes the catalytic subunit p110α of class IA PI3K. We corrected here, it not the gene, it is about PI3K inhibitor, lines: 291-293.
  14. Angiogeensis’ was corrected angiogenesis, line 921.

We are hopeful that the quality of the manuscript has been improved and that we fulfilled your requirements. Thank you very much for taking into consideration the publishing of our manuscript. 

Kind regards,

Lecturer Constantin Stefani

“Carol Davila” University of Medicine and Pharmacy

constantin.stefani@umfcd.ro

Lecturer Daniela Miricescu

“Carol Davila” University of Medicine and Pharmacy

daniela.miricescu@umfcd.ro

Reviewer 2 Report

Comments and Suggestions for Authors

January 15, 2024

Ms. Ref. No.: ijms-2811686

Journal: International Journal of Molecular Sciences.

Title: Targeting PI3K/AKT mTOR and MAPK signaling pathways in Gastric Cancer.

Comments:

Thank you for your efforts in writing this review article on a very pertinent topic. I have some observations where mentioned in the following paragraphs that will be useful for its improvement:

1-      According to some references, males are twice as likely to develop gastric cancer as females. It has been suggested that estrogen may protect women from developing this type of cancer. Is there a relationship between estrogen and the results of this study?

2-      The manuscript under consideration has been found to cite a total of 221 articles in its reference section. However, the criteria for the inclusion and exclusion of these articles remain unclear. It should be better to assess the quality and reliability of the research findings presented in the manuscript, it is crucial to establish the criteria utilized for selecting the cited references. Therefore, a detailed account of the criteria used for inclusion and exclusion of references would be highly appreciated.

3-      Could you please elaborate on the methodology employed to establish the time frame for this study? Your response would be of immense help in ensuring that all prerequisites are met.

4-      Approximately 10% of instances are present in family lines, while genetic syndromes, such as hereditary diffuse gastric cancer, account for 1% to 3% of cases. Can you explain the correlation between these reasons and the information provided in the article?

5-      To enhance the introduction's readability, it is advisable to consider incorporating some of the following references:

·         https://doi.org/10.3390/ijms25020999

·         https://doi.org/10.1007/s12013-023-01171-y

·         https://doi.org/10.3390/ijms25010252

Author Response

Dear reviewer,

On behalf of our research group, we would like to thank you for your time and your comments. We highly appreciated your recommendations and taking into consideration your suggestions, we made the following changes to the manuscript:

  1. The cited article does not specify the role of estrogen in the pathogenesis of gastric cancer. But we consider that this question is very important, so we introduced a phrase into the article to specify this aspect. Lines: 57-58.
  2. We wrote a narrative review, not a systematic review, therefore there were no specific research inclusion and exclusion criteria, we only had a topic of interest while researching the references. "Narrative reviews have no predetermined research question or specified search strategy, only a topic of interest. They are not systematic and follow no specified protocol" - Demiris, G., Oliver, D. P., & Washington, K. T. (2019). Defining and analyzing the problem. Behavioral intervention research in hospice and palliative care: Building an evidence base, 27-39.
  3. We considered that the most important aspects to understand better the molecular aspect of this malignant pathology is to explore the risk factors, to describe the most important two signaling pathways that are dysregulated and also to observe how many clinical studies are used for PI3K/AKT/mTOR  and MAPK inhibitors.

  1. Regarding the genetic syndromes and GC, we introduce a paragraph, lines: 245-250.

  1. The suggested references were inserted as references number: 5, 10, 32.

Kind regards,

Lecturer Constantin Stefani

“Carol Davila” University of Medicine and Pharmacy

constantin.stefani@umfcd.ro

Lecturer Daniela Miricescu

“Carol Davila” University of Medicine and Pharmacy

 daniela.miricescu@umfcd.ro

Reviewer 3 Report

Comments and Suggestions for Authors

The manuscript “Targeting PI3K/AKT/mTOR and MAPK signaling pathways in Gastric Cancer” by Morgos et al. is a comprehensive review of the involvement of two very crucial cell signaling pathways – PI3K/AKT/mTOR and MAOK signaling pathways in Gastric Cancer (GC) pathogenesis. Authors introduced GC and discussed the risk factors associated with GC pathophysiology. Authors gave an overview of the two signaling pathways that are of interest of this review and then discussed how different components of these pathways are involved in different stages of GC pathology. Authors also discussed how different inhibitors of these pathways have been shown to have the potential to act as antitumor agents.

The aim of this review manuscript is important and falls within the scope of the journal. However, the manuscript is very unorganized and is just like a compilation of all the relevant (and some irrelevant) literature! A good flow is absolutely missing! There are many unrelated/irrelevant pieces of information throughout the manuscript and some of those are listed below. Authors should realize that just putting hundreds of references in one place doesn’t make it a scientific review. The authors’ insight is largely missing. A good review article contains authors’ perspective on a scientific topic and some discussion on the future goals. Section 2 and section 6 needs to be shortened. The bibliography is unnecessarily long, and irrelevant or similar references need to be discarded. Section 5 is completely unorganized where all the information is just dumped.

Specific comments:

1.      The discussion on risk factors that contribute to Gastric Cancer (GC) pathology is unnecessarily long and that is not the theme of this review. This section needs to be shortened.

2.      A pictorial presentation of the overview of PI3K/AKT/mTOR and MAPK signaling pathway would be very useful.

3.      Figure 1. I am having trouble to read the words in the flow chart. Please consider to increase the font size.

4.      Line 243-247. The meaning of this sentence is not clear.

5.      Line 250: ‘actives’ should be ‘activates’

6.      Line 252: ‘PH’ is an abbreviation of ‘pleckstrin homology’

7.      Line 355-56. “Installation of metastasis will induce cancer treatment failure and an increased risk of mortality”. What does this sentence mean?

8.      Line 392-396. How this piece of information is relevant here? The authors were discussing the contribution of FGFR2 and suddenly a sentence appeared that talks about the impact of H. pylori infection on various cellular processes. This kind of mistakes spoils the enthusiasm of the readers!

9.      Line 396-397. Again, how this sentence is relevant here?

10.   Line 398-400. DDIT4 inhibits mTORC1 by stabilizing the TSC1/TSC2 complex. Very well, but where is any connection with GC? 

11.   Line 431. The gene name is CSNK2A1 not CNK2A1.

12.   Line 431. The authors mentioned that the role of CSNK2A1 in GC remains unclear. However, the article they cited (reference 176) revealed that CSNK2A1 promotes GC invasion through PI3K/AKT/mTOR pathway.

13.   Section 6 – “PI3K/AKT/mTOR and MAPK inhibitors” is unreasonably long. Authors don’t need to discuss detail methodology of how different inhibitors were tested in clinical trials by different groups but should be more concise.

14.   Ref. 205. Zhang J and his co-workers observed that rapamycin was able to reverse the cancer-promoting activity of NUP37. This is not a convincing data to include rapamycin in the list. Authors should rather include Yao et al. 2011 Hepatogastroenterology article that clearly shows involvement of rapamycin in GC.

15.   A slash “/” symbol is missing between AKT and mTOR in the title of the manuscript.

16.   There are many typographical errors and incomplete sentences or sentences without any meaning throughout the manuscript. The whole manuscript needs to be re-written preferably by a native English speaker.

Comments on the Quality of English Language

Quality of English needs to be improved. 

Author Response

Dear reviewer,

On behalf of our research group, we would like to thank you for your time and your comments. We highly appreciated your recommendations and taking into consideration your suggestions, we made the following changes to the manuscript:

  1. Figure 1 has become figure 2 and the font size was increased, line 442.
  2. New figure to present PI3K/AKT/mTOR and MAPK pathways: line 324.
  3. The sentence was re-written, lines: 307-311.
  4. The word “activates”, was corrected, lines: 314-315.
  5. Full name for PH, was added, line 317.
  6. The phrase: “Installation of metastasis will induce cancer treatment failure and an increased risk of mortality” was removed from the manuscript, lines: 459-460.
  7. The information about FGFR2 and pylorifrom section 5 was removed, lines: 496-503.
  8. We removed the phrase from lines 502-503.
  9. DDIT4 inhibits mTORC1 by stabilizing the TSC1/TSC2 complex. Very well, but where is any connection with GC? - The phrase was removed, lines: 504-511.
  10. We corrected the gene name, and also specify the role of CSNK2A1 in CG pathogenesis, lines: 534-540.
  11. We performed modifications in section 6, and removed details regarding the studies, lines: 562-565; 570-577; 558-600; 602-603; 676-679; 701-717; 761-762; 783-785; 914-915; 921-923.
  12. We changed the reference 205 with the specified one: line 783.
  13. We added "/" symbol in the title of the manuscript, line 2.
  14. We shorted section 2, lines: 112-113; 144-151, 245, 257.
  15. We shorted section 5, lines: 469-471; 482-490; 504-511.
  16. We corrected the English errors from the manuscript.

We are hopeful that the quality of the manuscript has been improved and that we fulfilled your requirements. Thank you very much for taking into consideration the publishing of our manuscript. 

Kind regards,

Lecturer Constantin Stefani

“Carol Davila” University of Medicine and Pharmacy

constantin.stefani@umfcd.ro

Lecturer Daniela Miricescu

“Carol Davila” University of Medicine and Pharmacy

daniela.miricescu@umfcd.ro

Reviewer 4 Report

Comments and Suggestions for Authors

This study reviews the PI3K/AKT mTOR and MAPK signaling pathways in gastric cancer. These pathways are very important to understand the malignancy in gastric cancer.

The titles of 1. Introduction and 2. Risk factors are not quite specific. Authors may try to have sub-sections of "incidence of gastric cancer" and "classification of gastric cancer" under the 1. Introduction. 

6. PI3K/AKT/mTOR and MAPK inhibitors can also have sub-sections such as "everolimus", "PI3Kbeta inhibitors", "PI3K/mTOR inhibitors", etc. The subsections may improve the manuscripts to be more readable. 

Author Response

Dear reviewer,

On behalf of our research group, we would like to thank you for your time and your comments. We highly appreciated your recommendations and taking into consideration your suggestions, we made the following changes to the manuscript:

The titles of 1. Introduction and 2. Risk factors are not quite specific. Authors may try to have sub-sections of "incidence of gastric cancer" and "classification of gastric cancer" under the 1. Introduction.

We split the first section Introduction into:

  • Gastric cancer incidence: line 48
  • Gastric cancer classification: line 75

The second section about risk factors were also split as follows:

2.1. Helicobacter pylori, line: 109

2.2 Diet, line: 151

2.3 Systemic diseases, line: 218

2.5 Genetic factors, line: 244

PI3K/AKT/mTOR and MAPK inhibitors can also have sub-sections such as "everolimus", "PI3Kbeta inhibitors", "PI3K/mTOR inhibitors", etc. The subsections may improve the manuscripts to be more readable. PI3K/AKT/mTOR and MAPK inhibitors can also have sub-sections such as "everolimus", "PI3Kbeta inhibitors", "PI3K/mTOR inhibitors", etc.

Regarding this section, it was a little difficult to divide according to a specific class of inhibitor, because usually one inhibitor was used together with other inhibitors. We divided this section as follows:

6.1 Inhibitors used in clinical trials, line: 553.

6.2 Inhibitors used in in vitro and in vivo studies, line: 721.

We are hopeful that the quality of the manuscript has been improved and that we fulfilled your requirements. Thank you very much for taking into consideration the publishing of our manuscript. 

Kind regards,

Lecturer Constantin Stefani

“Carol Davila” University of Medicine and Pharmacy

constantin.stefani@umfcd.ro

Lecturer Daniela Miricescu

“Carol Davila” University of Medicine and Pharmacy

daniela.miricescu@umfcd.ro

Round 2

Reviewer 1 Report

Comments and Suggestions for Authors

All the points that were raised have been addressed, and not only have they been corrected, but the overall paper writing has been significantly improved. Therefore, I recommend accepting it as it is.

Reviewer 3 Report

Comments and Suggestions for Authors

Authors have addressed the concerns raised by me to the best of their capacity and revised the manuscript accordingly. The revised manuscript can be accepeted for publication.